# Circular RNAs in the Origin of Developmental Lung Disease: Promising Diagnostic and Therapeutic Biomarkers

**DOI:** 10.3390/biom13030533

**Published:** 2023-03-15

**Authors:** Yajie Tong, Shuqing Zhang, Suzette Riddle, Rui Song, Dongmei Yue

**Affiliations:** 1Department of Pediatrics, Shengjing Hospital of China Medical University, Shenyang 110004, China; 2School of Pharmacy, China Medical University, Shenyang 110122, China; 3Cardiovascular Pulmonary Research Laboratories, Departments of Pediatrics and Medicine, University of Colorado Anschutz Medical Campus, Aurora, CO 80045, USA; 4Lawrence D. Longo MD Center for Perinatal Biology, Department of Basic Sciences, Loma Linda University School of Medicine, Loma Linda, CA 92350, USA

**Keywords:** circular RNA, miRNA, lung, developmental disorder, placenta

## Abstract

Circular RNA (circRNA) is a newly discovered noncoding RNA that regulates gene transcription, binds to RNA-related proteins, and encodes protein microRNAs (miRNAs). The development of molecular biomarkers such as circRNAs holds great promise in the diagnosis and prognosis of clinical disorders. Importantly, circRNA-mediated maternal-fetus risk factors including environmental (high altitude), maternal (preeclampsia, smoking, and chorioamnionitis), placental, and fetal (preterm birth and low birth weight) factors are the early origins and likely to contribute to the occurrence and progression of developmental and pediatric cardiopulmonary disorders. Although studies of circRNAs in normal cardiopulmonary development and developmental diseases have just begun, some studies have revealed their expression patterns. Here, we provide an overview of circRNAs’ biogenesis and biological functions. Furthermore, this review aims to emphasize the importance of circRNAs in maternal-fetus risk factors. Likewise, the potential biomarker and therapeutic target of circRNAs in developmental and pediatric lung diseases are explored.

## 1. Introduction

Circular RNA (circRNA) was first described in RNA viruses/viroids as an error of endogenous RNA junctions [1]. Initially, the function of circRNA was thought to be restricted only to the region of Y chromosome involved in sex determination [2]. In recent years, however, sequencing techniques and computational analyses have revealed that circRNAs have diverse functions [3]. Despite these advances, many questions remain unanswered, such as how microRNAs or RNA binding proteins interact with circRNA, and how circRNAs act as a scaffold to modulate protein complex formation.

The lack of a 3′ and 5′ end in circRNA prevents it from being digested by ribonucleases such as RNase R. Hence, circRNA has a half-life ten times that of linear RNA. CircRNA sequences are highly conserved [4]. CircRNAs are widely found in animal and human cells and play a crucial role in gene transcription and post-transcriptional gene expression. CircRNA can function as an RNA transporter, protein binder, transcription factor, regulatory factor, and miRNA sponge [5,6,7,8]. Circular RNAs contain miRNA response elements, which bind to miRNAs and compete for miRNA-binding sites. Consequently, circRNAs function as intracellular competitive endogenous RNA (ceRNA) by antagonizing miRNA function, which plays a key role in lung development and disease.

Currently over 20,000 circRNAs have been identified. In general, circRNAs are named based on their parental genes or specific functions. For instance, cerebellar degeneration-related protein antisense RNA (CDR1as) is also called ciRS-7 (circRNA sponge for miR-7) [9,10]. In mammals, circRNAs play an essential role in many tissues, including brain, blood, heart, liver, kidney, placenta, and lung [11]. Clinically, the contribution of aberrant circRNA expression to the pathogenesis of cardiopulmonary injuries, as well as the potential of targeting circRNA-miRNA associations is being pursued. (Clinical trials: NCT04864457, NCT03170830, NCT03766204). In addition to adult tissues, circular RNA has been discovered in developing tissues [11,12,13]. In recent years, circRNAs have emerged as important regulatory molecules for cardiovascular and pulmonary development, as well as diseases of aberrant development [11,12,14,15,16,17,18,19]. Recent research has proven that genetic factors play an integral role in the progression of developmental lung diseases such as bronchopulmonary dysplasia (BPD) [7,20] and neonatal acute respiratory distress syndrome (ARDS). Preterm birth can profoundly affect and delay cardiopulmonary development and transition contributing to BPD and neonatal ARDS [21,22].

Remarkable advances have been achieved in circRNA biology over the past few decades, but the effect of regulatory networks on circRNA function and regulation remain largely unexplored in health and disease. Here, we briefly overview circRNA biogenesis and their mechanisms of action in general and important analytic approaches of circRNA profiling. Additionally, we will discuss the latest data regarding their role in mesendoderm differentiation, pulmonary bud formation, lung branching morphogenesis, and vascularization relevant to pulmonary development and related diseases. Finally, circRNA analysis in diagnostic and therapeutic contexts are discussed.

## 2. Biogenesis and Biological Functions of circRNAs

mRNA precursers (pre-mRNA) typically include one to five exons and introns. Together, the combined sequence of both exons and introns can be as much as three times longer than linear processed mRNA alone [23]. CircRNA appears to originate from pre-mRNA, forming a single-stranded RNA circle with covalently joined 5′ and 3′ ends [3,23]. Three types of circRNAs can be classified according to their sequences: intronic circRNA (CiRNA), extron-intron circRNA (ElcRNA), and exonic circRNA (EcRNA) [24,25]. CircRNAs with exonic sequences are exported to the cytoplasm, whereas circRNAs with introns are anchored in the nucleus. To date, the majority of discovered circRNAs display a predominantly exonic structure and are primarily located in the cytoplasm [26].

CircRNAs usually are composed of exons 2 and 3 of the gene locus. Most circular RNAs are generated at the expense of their linear counterparts since they are derived from constitutive exons. A large portion of circRNAs contain the canonical splice site motif, suggesting that the canonical spliceosome is involved. When transcription termination factors (for example, SF3b or SF3a) or core spliceosome components (for example, SF3b) are inhibited, circRNAs can become the dominant RNA transcript. Transcripts that are read through can then be backspliced into downstream genes. CircRNAs can also be produced by exon skipping, through either the lariat-guided or intron-guided methods [27]. Currently, there are three models for exon splicing during circRNA formation: exon skipping backsplicing, intron paring backsplicing, and RNA binding proteins (RBP)-driven backsplicing [2,28,29] (Figure 1). The primary working hypothesis for formation of circRNA is that looping the intron sequences along the upstream junction site brings these two sites close together allowing a phosphodiester bond junction to occur where the 3′ and 5′ sites are joined [30]. This complex can be mediated by specific motifs at the 5′ and 3′ sides of introns (exon skipping backsplicing), base pairing between inverted repeating elements (intron paring driven backsplicing), or by dimerization with RNA binding proteins. When exons are skipped, they form a giant lariat, which undergoes internal cleavage, removing the intron and generating ElcRNA and EcRNA [31]. Both intron-pairing circularization or lariat intron driven ciRNA generation involve classic spliceosomes to generate circRNA [29]. Additionally, it has been found that when lariat introns do not detach from their parent mRNA at the level of the branch point site, ciRNAs are formed after trimming the lariat tail [32] (Figure 1).

As research continues, more will be learned regarding the regulation of splicing and circularization to generate circRNAs. CircRNA biogenesis can contain the 5′ end of the pre-mRNA upstream exon coupled to the downstream exon at the 3′ end. CircRNAs have covalently closed loops with no 3′ cap and no 5′ end. Therefore, their resistance to fluid absorption and ribonuclease (RNase) degradation leads to the relative stability of circRNA in the body compared to linear RNA [33,34]. The back-splicing of circRNA is dependent on DNA sequences (complementary intronic (cis)-elements sequences) and RNA-binding proteins (trans-factors) [33,35]. Circular RNAs are almost exclusively exonic and lack intron segregation. In fact, it was found that it is not exon sequences but circRNA’s complementary introns that regulate circRNA synthesis [28,36]. Moreover, complementary side sequences are enriched with circRNA introns in various species such as Caenorhabditis elegans, rodent, pig, and human [37,38,39,40].

Several mechanisms have been discovered for regulating circRNA biogenesis. Inhibiting spliceosomes by depleting U2 small nuclear ribonucleoprotein components, can increase circular to linear RNA ratios [41]. Alternative pathways can direct the newly formed RNA to a pathway that boosts backsplicing when pre-mRNA processing slows down. RNA binding proteins (RBPs) such as heterogeneous nuclear ribonucleoprotein (hnRNP), serine-arginine (SR), and FUS protein [42,43], as well as splicing regulators such as NF110 and NF90, Muscleblind, and NOVA2 can bind to intronic sequences flanking circularized exons and stabilize CIS pairs enhancing the production of circRNA [44,45,46]. Conversely, circRNA formation is inhibited by some RBPs. As an example, by binding to reverse complementary ALU elements, DEAH-box helicase 9 (DHX9) influences the uncoiling of Alu (*Arthrobacter luteus*) elements and inhibits the formation of circRNA [47]. Alu elements, transposable segments of DNA that are recognized by the Alu endonuclease restriction enzyme, are found throughout the genome. Originally thought to be parasitic DNA, evolutionary and functional roles are being discovered for this large family of retrotransposons. Among the newly discovered roles, complementary Alu segments in introns facilitate a more accessible junction leading to more circRNA generation [48]. In addition, when other factors such as the pre-mRNA 3′ end processing endonuclease Cpsf73 are depleted, each one may cumulatively disrupt circRNA formation [41]. ATP-dependent RNA helicase A and endogenous double-stranded RNA are required for the biogenesis of circRNAs that rely on base pairing between reverse repeats [44].

### 2.1. Transcriptional Regulation

In the nucleus, circRNAs such as EIciRNAs can directly bind to elongated RNA Pol II binding sites or interact with the Pol II transcription complex after forming EIciRNA–U1 snRNP complexes through RNA–RNA interactions. Both interactions regulate the transcription of circRNA parent genes (Figure 2). In human cells, ci-ankrd52 activates RNA polymerase II and enhances transcription of the parent gene ANKRD52 [49]. Binding of CircEIF3J and circPAIP2 to U1 snRNA through the U1-binding site in EIciRNA was demonstrated to be required for the transcription-enhancing effect of these two EIciRNAs [50]. In addition, some circRNAs such as circITGA7, circ-HuR, circ-STAT3, and circ-DAB1 regulate the parent genes’ transcription via their interaction and modulation of transcriptional factors [51,52,53,54]. CircRNAs, such as FECR1 and TAH-circRNAs, induced DNA methylation and thus contributed to circRNA-mediated transcriptional regulation of parent genes [55,56].

### 2.2. CircRNA as Super Sponge for miRNA

miRNA is noncoding RNA approximately 18 to 25 nucleotides long and an essential epigenetic regulator in eukaryotes. Mature miRNA can directly bind to target mRNAs, resulting in degradation of target mRNAs or suppression of target mRNA’s translation in lung development and diseases [57,58,59]. Remarkably, it has been verified that circRNAs contain miRNA response elements (MER), which can competitively bind to miRNAs [32] (Figure 2). Therefore, circRNAs act as intracellular competitive endogenous RNA (ceRNA) to antagonize miRNA function. The powerful miRNA sponge function of circRNAs was verified for the first time in 2013 [10]. Circular RNA, ciRS-7 (also termed CDR1as), was identified and co-expressed with miR-7 in the mouse brain. Significantly, the combination of ciRS-7 and miR-7 contributed to increased miR-7 target gene expression through suppressed miR-7 activity. This supports ciRS-7 as a miR-7 inhibitor, attributable to the more than 70 conventional binding sites for miR-7 on ciRS-7 [10]. At the same time, the testis-specific circRNA, sex-determining region Y (Sry)9, was found to have 16 putative target sites for miR-138 and was demonstrated to interact with miR-138, leading to decreased knockdown potential for miR-138 on its target genes [10]. Increasing evidence has accumulated for miRNA sponge effects by circRNA as a general phenomenon in the normal development and progression of various disease.

### 2.3. CircRNA as RNA Binding Protein Sponge

RNA binding proteins (RBPs) can target RNA through RNA-binding domains to regulate gene expression in different cellular processes such as cell morphology and differentiation, cell proliferation, response to oxidative stress, aging, and apoptosis [60]. CircRNAs have been found to play an essential role in forming this stable RNA protein complex that influences host gene expression [35] (Figure 2). In particular, circ-Mbl, circ-PAIP2, and circ-EIF3J interact with RNA polymerase II to enhance parent gene expression [41,50,61]. CircE2F2 or circPABPN1 interacts with HuR to regulate the stability and translation of target E2F2 or PABPN1mRNA [62]. CircCUX1, through its interaction with EWS RNA binding protein 1, facilitates MAZ-mediated CUX1 transcription [63]. In addition, circRNAs can sequester RBPs in order to affect the translocation of these proteins, which then affects the RBP-target gene regulation [64,65]. To be noted, circRNAs’ dynamic tertiary structure may be affected by different cell types, tissues, and developmental stages, which can affect their ability to bind to various proteins. Circular RNAs can display a variety of functions as a result of different circRNA–RBP interactions [66].

### 2.4. CircRNA as mRNA for Protein Coding

Several aspects of circRNA sequence have raised the possibility of translation. Several in vitro and in vivo studies have shown that circular RNA have an intrinsic entry site (IRES) or a modification of N6-methyladenosine (m6A) that can facilitate translation into peptides (Figure 2) [67,68,69]. Approximately 13% of all discovered circRNA sequences contain m6A consensus motifs. The YTHDF3 reader can recruit translation initiation factors to begin protein translation at a single m6A site and initiate circRNA translation. m6A modifications are commonly distributed in circRNAs and can strongly influence the efficiency of circRNA translation [70]. Many circRNAs such as circZNF609, circPINTexon2, circFBXW7, circSHPRH, circ-AKT3 and circβ-catenin contain the initial codon of ribosome-associated mRNAs [71,72,73,74,75,76]. Peptides can be translated from the small open reading frames (sORFs) of these circRNAs using coupling-dependent and non-constrained mechanisms, but not cap-dependent mechanisms since circRNAs lack a 5′cap [67].

Pamudurti et al. in their study presented evidence supporting circRNA translation: 1. circRNA was associated explicitly with translation ribosome; 2. proteins were generated from circRNA minigenes; 3. ribosome footprint readings were supported by the stop codon in circMbl; 4. sequences were identified that promoted cap-independent translation on several circRNAs; 5. a novel isoform of a peptide was detected coinciding with circMbl. However, circRNAs associated with ribosomes were detected in lower abundance than free circRNAs and some minigenes did not lead to protein production, suggesting that a specific process promotes translation of a subset of circRNAs [77].

### 2.5. Post-Translation Regulation

Recent studies revealed that circRNA could compete with enzymes and influence the post-translational modifications of full-length proteins coded by parent genes. Circβ-catenin-coding β-catenin-370aa competitively interacted with GSK3β and inhibited its binding to β-catenin, leading to the antagonization of GSK3β-induced β-catenin degradation [76]. AKT3-174aa encoded by circ-AKT3 competed with this AKT isotype for binding to pPDK1 and reduced AKT-Thr308 phosphorylation, indicating that AKT3-174aa plays a negative regulatory role in regulating phosphatidylinositol 3-kinase (PI3K)/AKT signaling activity [75]. Alternately, circFBXW7 was observed to interact with the deubiquitinating enzyme USP28, preventing USP28 from binding to FBXW7 and thus upregulating the expression of FBXW7 [78].

## 3. CircRNA and Risk Factors of Developmental and Pediatric Lung Diseases

Developmental and pediatric lung diseases are disorders of lung development affecting neonates, infants, children, and adolescents. The risk factors include environmental (high altitude), maternal (preeclampsia, smoking, and chorioamnionitis), placental, and fetal (preterm birth and low birth weight) factors. These risk factors are interrelated and interact during fetal development. They impact perinatal “plasticity”, lung development, as well as lung injury and repair processes. There is an increasing body of evidence that abnormal circRNA expression is associated with developmental diseases in humans and is associated with gene dysregulation [13,79].

In one animal model used to study the effects of high altitude (hypoxic) environmental stress and adaptation, circRNAs and mRNAs (SKIV2L2, PRKCSH, NewGene.10854.1, POR and LOC102286089) were found to be downregulated in yak lungs exposed to high altitude [80]. This study revealed the role of circRNAs in transcriptional changes in response to high altitude adaptation. In a human subject study, Qian et al. examined circRNA expression patterns in the placental tissues of pregnant women with preeclampsia and pregnant women who delivered healthily but prematurely [81]. They discovered significant differential expression of circRNAs in the placental tissues of women with preeclampsia: 143 circRNAs were upregulated, and 158 were downregulated [81]. In another study, 300 circRNAs were identified as differentially expressed between preeclampsia and normal placental tissues [82]. Among them, hg38_circ_0014736 and hsa_circ_0015382 were validated as significantly upregulated and hsa_circ_0007121 was significantly downregulated [82]. Remarkedly, the GEO databases GSE102897 and GSE96985 show different expression profiles of circRNAs in human preeclampsia and normal placentas. It has been shown that increased circRNAs such as circ_0001687, circLRRK1, circ_0008726, circ_0085296, circ_0011460, circ_0026552 and decreased circRNAs such as circ_0001513 play a role in trophoblast proliferation, migration, and angiogenesis during preeclampsia development [83,84,85,86,87,88].

Preeclampsia is one of the leading causes of premature birth. As a result of premature birth, there are a number of complications that result in significant morbidity and mortality, including bronchopulmonary dysplasia (BPD), hyaline membrane disease, and pulmonary interstitial emphysema [89]. A recent study has documented a circRNA expression profile associated with preterm birth, and preliminarily analyzed its regulatory mechanism and predictive value for preterm birth [90]. About 211 abnormally expressed circRNAs existed in the peripheral blood of preterm women. Among them, the top 20 circRNAs, including hsa-SCARF1_0001, hsa-GCN1_0003, hsa-RAD54L2_002, hsa-CREBBP_0001, hsa-FAM13B_0019, hsa-NUSAP1_0010, hsa-YY1AP1_0001, hsa-MORC3_0001, and hsa-RANBP9_0002, are related to immune/inflammatory pathways mediating the process of preterm birth [90]. Although there is not yet a clear and direct relationship between maternal-fetus risk factors and these disorders, these risk factors can affect prenatal lung development and can have a direct impact on neonatal health. Therefore, circRNA-mediated maternal-fetus risk factors are likely to contribute to the occurrence and progression of developmental and pediatric lung disorders (Figure 3).

Current research into the role of circRNAs in developmental and pediatric lung disorders is restricted to BPD. The current definition of BPD describes a developmental and pediatric lung disease with less heterogeneity, simplified alveolar surface, and reduced and dysmorphic vascular bed [91]. In preterm infants, placental, environmental, and genetic insults may result in abnormal alveolarization and vascularization of the lungs, increasing the possibility of developing BPD. There is clear evidence that pulmonary vascular and alveolar development are interdependent processes, and a negative correlation between alveolar simplification and distal lung angiogenesis has been observed [92]. Although BPD treatment and diagnosis are progressing, many patients still suffer from lung damage, leading to long-term lung dysfunction [93,94]. Much is still unknown regarding the multi-faceted and complex pathological process of BPD [95]. Further clarification of the molecular mechanisms leading to BPD can suggest novel molecular targets for diagnosis and treatment.

## 4. Co-Expression Networks of CircRNA-miRNA in the Progression of Developmental and Pediatric Lung Diseases

Although non-coding RNAs (ncRNAs) were once considered a waste product, they are now recognized as molecules that regulate many lung disorders. Since circRNAs act as sponges in the human body for miRNAs and can alter the function of those miRNAs, presumably circRNA can influence the occurrence and progression of diseases through regulation of miRNAs [13] (Figure 3).

Recent discoveries revealing the profile of circRNA expression in preterm infants with BPD is beginning to elucidate the role of circRNA in BPD and its dysregulated biological processes. In the peripheral venous blood of neonates with BPD, 491 circRNAs were markedly altered [20]. Significantly increased were circ_FANCL, circ_0009256, circ_0003037, circ_0009983, circ_0003357, and circ_0003122, while significantly decreased were circ_0014932, circ_0015109, circ_0017811, circ_0020588, and circ_0015066. These altered circRNAs likely contribute to the complex signaling pathways and biological processes in BPD [20]. Individuals with moderate BPD displayed a significant increase in circ_FANCL in correlation with oxygen treatment [20]. In another study, circABCC-4 levels in peripheral blood from 31 preterm infants with BPD were significantly increased, and the significance of the increase was positively correlated with poor long-term outcomes [96]. Further, both in vitro and in vivo, circABCC-4 promotes apoptosis and inhibits cell proliferation, important interrupted processes in BPD development [96]. Therefore, there is a clear indication that an interconnected circRNA expression profile in peripheral blood in neonates has important implications in both the diagnosis and pathogenesis of BPD.

In recent years, there has been considerable interest in exploring ncRNAs and miRNAs in BPD. Tao et al. [97] illustrated the interaction between the gene, RNA imprinted and accumulated in nucleus (Rian), and miR-421 in BPD models. They determined that Rian was downregulated in hyperoxia-induced BPD, which induced inflammatory responses via targeting miR-421 and upregulating miR-421 expression. Using analysis of miRNA-circRNA co-expression networks in peripheral venous blood from neonates with BPD, it was determined that many miRNAs can bind to one circRNA. At the same time, one miRNA can also regulate different circRNAs functions (Table 1). For instance, upregulated circ_FANCL interacts with let-7, miR-196, miR-20a, miR-22, and miR-26a. This network can be expanded through let-7 which can regulate the TGF-β/RAS/HMGA2 pathways [20,98]. CircABCC4 has been found to be related to BPD from genetic screening; this circRNA was found to share the miRNA response element of miR-663a with PLA2G6, strongly indicating that there exists an axis between these three molecules [96]. Furthermore, a circABCC4/miR-663a/PLA2G6 network was associated with the severity of the development and clinicopathological features of BPD [96]. In vitro studies documented circABCC4 targeting and downregulation of miR-663a expression, which directly inhibited PLA2G6 expression. In rat, six circRNAs were identified and positively correlated with BPD, while seven other circRNAs were negatively correlated with BPD [99]. In addition, Wang et al. [7] identified 634 miRNAs and 1545 circRNAs in BPD mouse models. Further, they generated circRNA-miRNA co-expression networks for seven upregulated circRNAs (e.g., Chr8:11226466|11231468, Chr9:108218013|108218410, Chr8:127415570|127426753, Chr13:10386649|103897928, Chr2:160750963|160752574, Chr11:106868535|106875939 and Chr3:15411189|15542472) and three downregulated circRNAs (e.g., Chr1:85202140|85659862, Chr1:177096967|177109738 and Chr14:70256360|70267506) (Table 2). Among the miRNAs identified in this circRNA-miRNA co-expression network, some miRNAs such as the let-7 family, miR-141, miR-100, miR-181b, miR-503, miR-29a, miR-135b, and miR-17 have previously been identified in the development of BPD [7,100,101,102,103]. This suggests that circRNA-miRNA networks can influence genes in signaling pathways associated with the progression of BPD, both positively and negatively.

In a recent study, circRNA expression profiles were investigated in neonatal acute respiratory distress syndrome (ARDS) [111]. In the United States, neonatal ARDS is the most common cause of respiratory distress in premature infants. It is noteworthy that babies born with ARDS are likely to develop BPD. Prematurity and the low birth weight of the infant are the most significant risk factors for both. Additionally, maternal diabetes, hypoxia, and ischemia during pregnancy contribute to the risk [22]. The Montreaux definition is a consensus definition for neonatal ARDS that covers neonates from birth to 44 weeks postmenstrual age (4 weeks if the baby is born at term) [112]. Neonatal ARDS is characterized by inflammation of the lungs and catabolism of surfactant molecules, leading to pulmonary dysfunction in neonates [112]. Neonatal ARDS remains one of the leading causes of morbidity and mortality in preterm infants despite treatment advances, such as antenatal corticosteroids and surfactants. Physician-scientists are continuing to research the mechanisms of neonatal ARDS and search for new therapeutic targets. In the blood samples of newborns with neonatal ARDS, Zhou et al. discovered 741 circRNAs that were downregulated and 588 that were upregulated compared to those in normal newborn blood [111]. Based on bioinformatic analysis of the parental genes of differentially expressed circRNAs, these circRNAs could be involved in protein synthesis and metabolism in neonatal ARDS [111]. As an example, the hsa_circ_0005389 gene regulates the amino acid transporter SLC38A10 [111], which is involved in immune response, nascent protein synthesis, and cell survival under oxidative stress [113]. In neonatal ARDS, three upregulated circRNAs (hsa_circ_0005389, hsa_circ_0000367, hsa_circ_0059571) and two downregulated circRNAs (hsa_circ_0058495, hsa_circ_0006608) were found to interact with 25 miRNAs and 125 target genes [111]. These genes contribute to inflammation in the early stages of neonatal ARDS by synthesizing and secreting endocrine hormones, such as glucocorticoids, and affecting gene regulatory cascades, such as the stress-activated MAPK cascade [111]. As a result, circRNAs may offer new approaches to diagnose neonatal ARDS and decrease the inflammatory response in neonatal ARDS and related diseases, potentially offering a new therapeutic route.

## 5. Potential Biomarkers or Therapeutic Targets of circRNAs in Developmental and Pediatric Lung Diseases

There is growing evidence that exosome-noncoding RNAs likely contribute to lung disease. Research has demonstrated that exosomes isolated from tracheal aspirates of infants with severe BPD and bronchoalveolar lavage fluids from hyperoxia-treated newborn mice contain reduced miR-876 levels, a miRNA implicated in BPD development [114]. A large body of research on exosome-noncoding RNAs in lung disorders has focused on exosome-miRNAs and exosome-, long, noncoding RNA (lncRNAs), while very little research has been conducted on other noncoding RNAs species such as exosome-circRNAs. In addition to microRNAs and lncRNAs, circRNAs were identified in exosomes, thus the important role of exosome-bound circRNAs in developmental lung disorders cannot be ignored. In a recent study, circRNAs, long noncoding RNAs, and mRNAs were profiled in the umbilical cord blood of newborns with BPD, and 317 circRNAs, 104 long noncoding RNAs, and 135 mRNAs were found to be altered. Through bioinformatic analysis, several potential exosomal circRNA/lncRNA–miRNA–mRNA networks were identified with relevance to BPD pathogenesis [115]. However, our knowledge of the connection of circRNAs with exosomes remains limited compared to that of exosomal lncRNA or miRNA, especially in regards to diagnosis and treatment of developmental and pediatric lung diseases.

A substantial amount of research has been conducted in the field of cardiovascular diseases on stem cell exosome-based biomarkers, therapy strategies, and drug delivery [116]. The use of nanoparticles as a carrier to deliver RNA specifically and efficiently to target cells is a significant development towards therapeutic exosome engineering. Exosomal circRNAs have been extensively studied as therapeutic targets for cancer. For example, overexpression of exosomal circSHKBP1 may promote gastric cancer proliferation, migration, invasion, and angiogenesis, while knockdown of exosomal circSHKBP1 reduces lung metastatic tumor size and number [117]. Once the mechanism by which exosomal circRNAs or exosomal circRNA/miRNA disseminate through body fluid and their role in abnormal lung development and lung disorders has been determined, circRNA-exosome-targeted therapy may offer novel targets for therapeutic intervention for lung regeneration and prevention of the harmful effects of developmental lung diseases.

In fascinating investigations, circRNAs have been engineered to act as sequestering sponges for miRNAs associated with human diseases [118,119]. This makes circRNA a useful tool for the study of molecular biology and molecular medicine because the engineered circRNA (circmiRs) can be targeted to the nucleus as well as the cytoplasm. In recent work, Lavenniah et al. designed a circRNA sponge to target miR-132 and -212, known pro-hypertrophic miRs in the heart [119]. Artificial circRNA sponges successfully targeted the miR-212/132 family, were successfully delivered to cardiomyocytes in vivo, and successfully reduced left ventricular hypertrophy [119]. In light of this, the potential of engineered circRNAs as a future therapeutic in humans is promising. The implications of this are tremendous as it suggests that circRNA has great promise as a method of controlling developmental and pediatric lung disease progression. Nevertheless, the biological functions and mechanisms of circRNA-miRNA must be explored and verified further to be used as a clinical approach for developmental and pediatric lung diseases.

CircRNAs serve as scaffolds for proteins and miRNA sponges, and can affect translation, transcription, and degradation of specific mRNAs. There will likely be wide-ranging developments in the circulating RNA field in the coming years. CircRNA transport, localization, degradation, and biological function will be characterized in greater depth. Though newly emerging, circRNA has been identified as an essential player in lung development and developmental lung diseases. The role of circRNAs as miRNA sponges in the normal development of lung tissue and lung diseases is still being explored, and their precise mechanism of action is still unknown. It is, therefore, necessary to perform more analyses on samples collected from humans to identify circRNAs involved in cardiopulmonary development and related diseases. Further discovery of circRNA and circRNA-miRNA networks offer promising targets for therapies for developmental and pediatric cardiopulmonary disorders. As circRNAs are identified as potential biomarkers for developmental lung disease, more in-depth exploration is needed to demonstrate their relative accuracy and reliability in BPD as well as other developmental and pediatric lung diseases. Much is still unknown regarding the molecular function of circRNA in lung development and diseases. With our increased understanding of circRNA mechanisms, we expect increased opportunities for innovative treatments targeting the many circRNA-based roles in physiological and pathological processes, including those of the developing lung.

## Figures and Tables

**Figure 1 biomolecules-13-00533-f001:**
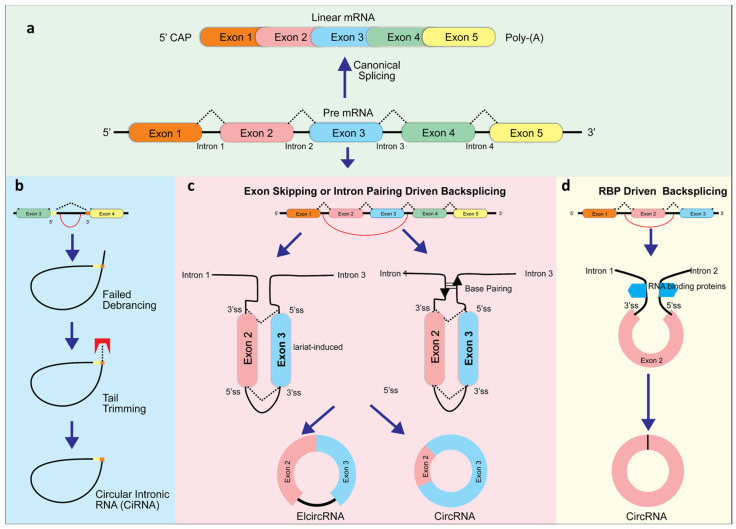
CircRNA biogenesis. The classic splicing typically generates linear RNA. However, this splicing can also produce circular intronic RNAs (ciRNAs) from intronic lariat precursors (**a**) when avoiding debranching via the presence of consensus RNA sequences (yellow) (**b**). In addition, long flanking introns, inverted repeat elements, and RNA binding proteins (RBP) boost backsplicing, including exon skipping (**c**), intron pairing driven (**c**), and RBP-driven backsplicing (**d**).

**Figure 2 biomolecules-13-00533-f002:**
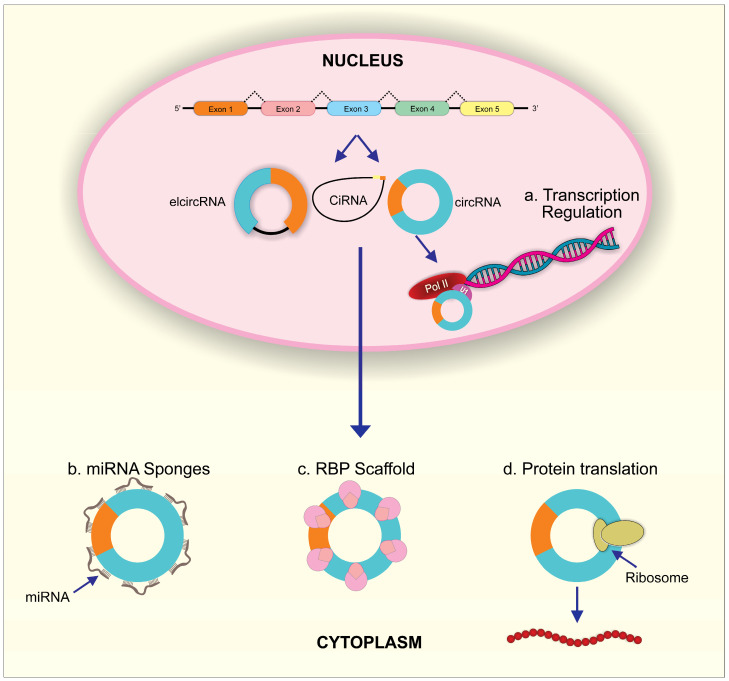
The biological functions of circRNAs. (**a**) In the nucleus, circRNAs can regulate parent RNA transcription through combining with U1 snRNP and then interacting with Pol II. In the cytoplasm, (**b**) circRNAs can act as miRNA sponges to inhibit miRNA activity; (**c**) circRNAs can scaffold RBPs to regulate post-transcriptional process; (**d**) circRNAs can initiate protein translation by recruiting ribosome.

**Figure 3 biomolecules-13-00533-f003:**
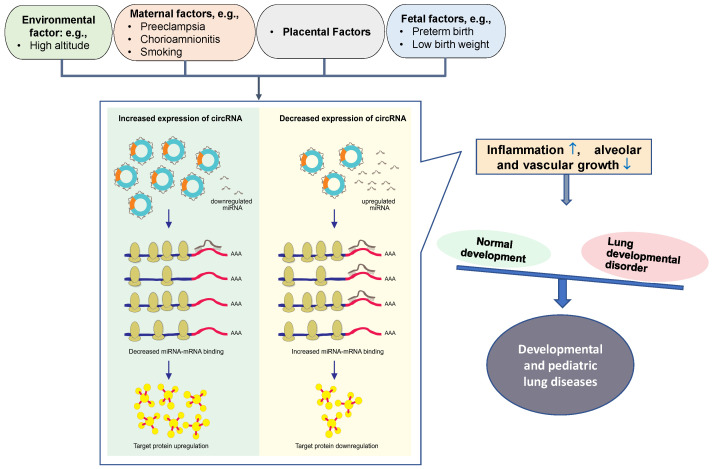
Schematic diagram of circRNA-miRNA interaction on controlling developmental and pediatric lung diseases. CircRNAs contain miRNA-binding sites. Maternal-fetus risk factors cause the alteration of circRNAs in placenta or/and circulation. Increased circRNAs can absorb more miRNAs, leading to miRNA downregulation; this reduces miRNA-mRNA binding, thereby upregulating miRNA target gene expression and protein translation. Conversely, decreased circRNAs and circRNA-miRNA interaction will downregulate miRNA target gene expression and protein translation. Eventually, the dysregulated proteins involved in the abnormal lung development can induce inflammation (
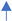
) and suppress the growth of alveoli and blood vessels (
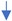
), potentially leading to developmental and pediatric lung diseases.

**Table 1 biomolecules-13-00533-t001:** CircRNAs-miRNA involved in lung development.

Species	CircRNA	Development Stages	Sponged miRNA	Potential Effect	References
Rat	circ:chr7: 24777879-24784993	pseudoglandular > canalicular > saccular > alveolar	miR-7a-2, miR-15a, miR-15b, miR-16, miR-93 *, miR-103, miR-107, miR-134, miR-190a, miR-193, miR-195, miR-196a, miR-218a-1-3p, miR-224, miR-322, miR-325, miR-326, miR-330, miR-448, miR-497, miR-500, miR-501, miR-582, miR-1199, miR-1297	VEGF expression and stem cell differentiation	[18,104]
	circ:chr3: 1988750-1998592	saccular > canalicular > pseudoglandular > alveolar	miR-15, miR-16, miR-96, miR-127, miR-130b, miR-140, miR-195, miR-201, miR-219b, miR-221 *, miR-322, miR-330, miR-412, miR-489, miR-497, miR-598, miR-615, miR-3551, miR-3580, miR-3584, miR-3593	Embryonic lung branching morphogenesis and epithelial cell fate	[18,105]
	circ:chr14:14620910-14624933	saccular > alveolar > canalicular > pseudoglandular	Let-7 family *, miR-15, miR-16, miR-30c, miR-30c, miR-103, miR-107, miR-143, miR-183, miR-185, miR-195, miR-322, miR-324, miR-362, miR-497, miR-500, miR-764, miR-3556, miR-3557, miR-3590, miR-6216, miR-6314, miR-6315	Lung bud formation, pulmonary branching morphogenesis, and lung epithelial cell proliferation	[18]
Mouse	circAlg12	pseudoglandular < canalicular < saccular < alveolar	miR-9, miR-141, miR-193, miR-188, miR-339	?	[17]
	circPtprm	pseudoglandular < canalicular < saccular < alveolar	miR-16, miR-130 *, miR-139, miR-143,	Lung airway and vascular morphogenesis	[17,105,106]
	circFilip1l	pseudoglandular < canalicular < saccular < alveolar	let-7 *, miR-15b, miR-34, miR-127, miR-148	Mesendoderm differentiation; lung bud formation, pulmonary branching morphogenesis, and lung epithelial cell proliferation	[17,107,108]
	circTtn	pseudoglandular > canalicular > saccular > alveolar	miR-17 *, miR-18, miR-20, miR-378b, miR-1943,	Embryonic lung epithelial branching morphogenesis	[17,109]
	circGalnt18	saccular > alveolar > canalicular > pseudoglandular	let-7 *, miR-17 *, miR-20, miR-92, miR-149	Lung bud formation, pulmonary branching morphogenesis, and lung epithelial cell proliferation	[17,108,109]
	circNcoa3	pseudoglandular < canalicular < saccular < alveolar	miR-17 *, miR-20, miR-106, miR-126, miR-138	Embryonic lung epithelial branching morphogenesis	[17,109]

* Verified in lung development. ? Unrevealed.

**Table 2 biomolecules-13-00533-t002:** Dysregulated circRNAs in BPD.

Species	CircRNA	Expression	Sponged miRNA	References
Human	circ_FANCL	up	let-7 *, miR-20a, miR-22, miR-26a, miR-196	[20,98]
	circ_0003122	up	?	[20]
	circ_0003357	up	?	[20]
	circ_0009983	up	?	[20]
	circ_0003037	up	?	[20]
	circ_0009256	up	?	[20]
	circ_0014932	down	?	[20]
	circ_0015109	down	?	[20]
	circ_0017811	down	?	[20]
	circ_0020588	down	?	[20]
	circ_0015066	down	?	[20]
Rat	chr9:52042894|52045136	up	?	[99]
	chr1:176334639|176346651	up	?	[99]
	chr8:46716191|46730691	up	?	[99]
	chr2:113322714|113345579	up	?	[99]
	chr20:18465901|18471939	up	?	[99]
	chr4:152526403|152529177	up	?	[99]
	chr1:198311923|198312195	down	?	[99]
	chr7:119987740|119990307	down	?	[99]
	chr9:66507281|66509028	down	?	[99]
	chr15:67588477|67588841	down	?	[99]
	chr19:9683854|9684128	down	?	[99]
	chr1:87010380|87010777	down	?	[99]
	chr8:71336875|71337745	down	?	[99]
Mouse	Chr8:11226466|11231468	up	let-7b *, miR-150	[7,17,107,108]
	Chr9:108218013|108218410	up	let-7b *, miR-22, miR-25, miR-34c, miR-93, miR-96, miR-297a, miR-383,	[7,17,107,108]
	Chr8:127415570|127426753	up	miR-21a, miR-25, miR-29b, miR-100 *, miR-107, miR-141 *, miR-181b *, miR-181d, miR-377, miR-511, miR-1943	[7,100]
	Chr13:10386649|103897928	up	let-7b *, let-7c *, miR-7b, miR-21a, miR-100 *, miR-107, miR-132, miR-141 *, miR-218, miR-181b *, miR-181d, miR-195a, miR-452, miR-503 *, miR-1943	[7,17,59,101,107,108,110]
	Chr2:160750963|160752574	up	miR-34c, miR-182, miR-489, miR-3473b	[7]
	Chr11:106868535|106875939	up	miR-96	[7]
	Chr3:15411189|15542472	up	miR-29a *, miR-106b, miR-132, miR-144, miR-195a, miR-199a, miR-218, miR-503 *, miR-669m, miR-1943	[7,59,101,110]
	Chr1:85202140|85659862	down	Let-7c *, miR-452	
	Chr1:177096967|177109738	down	let-7c *, miR-15a, miR-17 *, miR-21a, miR-22, miR-25, miR-33, miR-34a *, miR-92a, miR-93, miR-100 *, miR-106b, miR-107, miR-132, miR-135b *, miR-141 *, miR-181b *, miR-181d, miR-182, miR-195a, miR-196a, miR-199a, miR-200c, miR-218, miR-297a, miR-335, miR-362, miR-376b, miR-383, miR-449c, miR-503 *, miR-511, miR-669m, miR-1943	[7,59,101,110]
	Chr14:70256360|70267506	down	let-7b *, let-7c *, miR-15a, miR-20a, miR-93, miR-22, miR-144, miR-150, miR-153, miR-195a, miR-199a, miR-218, miR-322, miR-452, miR-503 *	[7,17,59,101,107,108,110]

* Verified in lung development and BPD. ? Unrevealed.

## Data Availability

Not applicable.

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
