# Peer review of "Circular RNAs in the Origin of Developmental Lung Disease: Promising Diagnostic and Therapeutic Biomarkers"

_biomolecules, 2023, doi:10.3390/biom13030533_

Round 1

Reviewer 1 Report

Summary of Paper:

This paper began with a detailed description of the origin and function of circRNAs in general and provided a detailed description of their three main functions (miRNA sponges, RBP scaffolds, and acting as mRNA for protein synthesis). The paper then explored the published literature on the role of different circRNAs in lung development and how dysregulation in circRNAs during lung development can lead to lung disease in newborns. The paper then discussed the more speculative future of using circRNA either in a therapeutic manner, i.e. to sequester “harmful” miRNAs, or in a diagnostic role to monitor potential problems in lung development, though the paper does make it clear these are only in the very early stages.

Summary of Critique and Recommendations:

Overall, the paper seemed to be grounded in good science. There does not seem to be any bias or self-promotion, and the authors did a good job of not being too hyperbolic in the speculation of the potential of circRNAs in therapeutics or diagnostics.  There is another review paper with a very similar title (A narrative review of circular RNAs as potential biomarkers and therapeutic targets for cardiovascular diseases) whose first half also describes the origin and function of circRNAs in a very similar manner. There are only so many ways to describe foundational things. However, the similarity is striking. Some of the sentences are a bit wordy as well when they list many circRNAs all in the same sentence, but I am also not sure of a way around this.

There are some issues in the figures that are listed below.

Comments and Recommendations:                  

·       Improve the quality of the graphics as, when they are viewed on the larger screen, they appear blurred.

·       Add more descriptive figure 2 legend.

·       Figure 3 should be moved to section 3, as that is where most of the content it depicts is. I know it is referenced once in section 2, but that could be reworked a little to put the figure in where it seems to fit best.

·       The arrow to the right in Figure 3 leading to the Pediatric lung disease bubble could give the impression that only decreased expression of circRNA leads to disease. Maybe make this a little more clear that both increased and decreased circRNA lead to disease.

·       Table 1 needs to be reworked to make things more clearly. Many cases of Sponged miRNAs run together, so it is difficult to tell which circRNA they correspond to. The reference they come from is also not always clear. 

·       Format table 2 so that the name of CirRNA is in one line. It is unclear if it is the name of the circRNA or the location e.g., chromosomal location. Feature either as a name or as a chromosomal location. I don’t know where are references to some of the examples. Both of the tables' layout is also confusing and hard to read. 

Author Response

We are greatly appreciative of the positive and constructive comments of the reviewers regarding our manuscript. Revisions have been made to the manuscript in response to Reviewers’ comments. We believe that the inclusion of reviewers’ comments has significantly further improved the manuscript.

Responses to Reviewer #1:

This paper began with a detailed description of the origin and function of circRNAs in general and provided a detailed description of their three main functions (miRNA sponges, RBP scaffolds, and acting as mRNA for protein synthesis). The paper then explored the published literature on the role of different circRNAs in lung development and how dysregulation in circRNAs during lung development can lead to lung disease in newborns. The paper then discussed the more speculative future of using circRNA either in a therapeutic manner, i.e. to sequester “harmful” miRNAs, or in a diagnostic role to monitor potential problems in lung development, though the paper does make it clear these are only in the very early stages.

Summary of Critique and Recommendations:

Comment:  Overall, the paper seemed to be grounded in good science. There does not seem to be any bias or self-promotion, and the authors did a good job of not being too hyperbolic in the speculation of the potential of circRNAs in therapeutics or diagnostics.  There is another review paper with a very similar title (A narrative review of circular RNAs as potential biomarkers and therapeutic targets for cardiovascular diseases) whose first half also describes the origin and function of circRNAs in a very similar manner. There are only so many ways to describe foundational things. However, the similarity is striking. Some of the sentences are a bit wordy as well when they list many circRNAs all in the same sentence, but I am also not sure of a way around this.

Response: We thank the Reviewer for the comments.  It is very true that there are only a limited number of ways to efficiently describe the extensive biological processes and functions of circRNA, thus leading to reviews on this topic following obviously similar patterns. In fact, our purpose in summarizing the current understanding of circRNA biological process and function was not to provide a novel review of circRNA biology, but to provide context for the remainder of our review, the current findings of circRNA research in the field of developmental lung disease and pathogenesis. To address this, we have changed the title of our paper to “Circular RNAs in the origins of developmental lung disease: Promising Diagnostic and Therapeutic Biomarkers”, so that its identity is less easily confused with the paper on circRNA in cardiovascular research.

          Unfortunately, at this early stage of circRNA research the names of the different molecules have yet to be formalized and are often rather long. We understand this can contribute to the wordiness of the writing but have not found a solution to this problem other than simply referring to tables which we feel limits the depth we provide to the reader by including specific names of the circRNAs within the sentences.

There are some issues in the figures that are listed below.

Comments and Recommendations:                 

Comment:  Improve the quality of the graphics as, when they are viewed on the larger screen, they appear blurred.

Response:  We have improved the quality of the Figures.

Comment: Add more descriptive figure 2 legend.

Response:  As suggested, we have added more additional information to the Figure 2 legend.

Comment: Figure 3 should be moved to section 3, as that is where most of the content it depicts is. I know it is referenced once in section 2, but that could be reworked a little to put the figure in where it seems to fit best.

Response:  As suggested, we have revised Figure 3 and moved Figure 3 to section 3.

Comment: The arrow to the right in Figure 3 leading to the Pediatric lung disease bubble could give the impression that only decreased expression of circRNA leads to disease. Maybe make this a little more clear that both increased and decreased circRNA lead to disease.

Response: As suggested, we have revised Figure 3.

Comment: Table 1 needs to be reworked to make things more clearly. Many cases of Sponged miRNAs run together, so it is difficult to tell which circRNA they correspond to. The reference they come from is also not always clear.

Response: We thank the Reviewer for the comment. We have modified the Table 1 to make it clearer and more readable. In each line of the Table 1, the sponged miRNAs are predicted to be the targets of one circRNA by bioinformatic analysis.

Comment:  Format table 2 so that the name of CirRNA is in one line. It is unclear if it is the name of the circRNA or the location e.g., chromosomal location. Feature either as a name or as a chromosomal location. I don’t know where are references to some of the examples. Both of the tables' layout is also confusing and hard to read.

Response:

We have modified the Table 2 to make it clearer and more readable. Despite the explosion of circRNA research recently, the need for a standard nomenclature is becoming increasingly evident for clear communications of circRNA research. The current terminology of circRNAs is very confusing for both bioinformatic and experimental research. In Circbase, circRNAs are named by some arbitrary number that does not provide any information about the parent gene or the location of the circRNA on the chromosome. According to some publications, circRNAs are also named according to their parent gene or chromosome location.

Reviewer 2 Report

This is a well-written review of circRNA in the early origins of lung disease. The authors summarize the circRNAs in lung development and organize them into a table, which is easy to follow. Some minor edits are needed:

1. Figures, tables, and references are not in alignment. The format needs to be revised.

2. In section 5, the authors could include more examples and studies on how people target, diagnose, and engineer circRNA in treating diseases.

Author Response

We are greatly appreciative of the positive and constructive comments of the reviewers regarding our manuscript. Revisions have been made to the manuscript in response to Reviewers’ comments. We believe that the inclusion of reviewers’ comments has significantly further improved the manuscript.

Responses to Reviewer #2:

This is a well-written review of circRNA in the early origins of lung disease. The authors summarize the circRNAs in lung development and organize them into a table, which is easy to follow. Some minor edits are needed:

Comment: 1. Figures, tables, and references are not in alignment. The format needs to be revised.

Response: We have revised the format of the Figures, Tables and References according to the Journal’s format requirement.

Comment: 2. In section 5, the authors could include more examples and studies on how people target, diagnose, and engineer circRNA in treating diseases.

Response: As suggested, we have included more studies to section 5. Lines 389-393 “In a recent study, circRNAs, long noncoding RNAs, and mRNAs were profiled in the umbilical cord blood of newborns with BPD, and 317 circRNAs, 104 long noncoding RNAs, and 135 mRNAs were found to be altered. Through bioinformatic analysis, several potential exosomal circRNA/lncRNA–miRNA–mRNA networks were identified with relevance to BPD pathogenesis [115].”; Lines 400-404 “Exosomal circRNAs have been extensively studied as therapeutic targets for cancer. For example, overexpression of exosomal circSHKBP1 may promote gastric cancer proliferation, migration, invasion, and angiogenesis, while knockdown of exosomal circSHKBP1 reduces lung metastatic tumor size and number [117].”

Reviewer 3 Report

I commend the authors on an excellent review of an emerging molecular mechanism of considerable scope. The writing is thorough, which covers the basic details and specifications of circRNAs, to their relationship with miRNAs, and to their plausibility as biomarkers/therapeutic targets.

I have a few minor comments:

Line 45 – authors could expand on the essential role in synaptic and neuronal functions.

Line 56 – consider rewording this line.

Intro – The behaviour of circRNAs acting as sponge for miRNAs can be covered and expanded upon first in the introduction.

Figure 1 – the lariat formation could be labelled in the figure

Line 112 – mention ribonuclease and the relative stability of circRNA in the body compared with RNA.

Line 118  - Seems sensible to reorder the species with humans coming last, i.e rodent, pig, human..

Line 120 – (snRNP) expand on first use

Line 181 – authors can reword this sentence to include something of the like “dysregulation, imbalance and abnormal lung development can induce…..”

Heading 2.3 – This topic seems to be of extra significance. Perhaps this paragraph can be expanded to include more information at hand.

Line 304 – expand Rian on first use.    

Author Response

We are greatly appreciative of the positive and constructive comments of the reviewers regarding our manuscript. Revisions have been made to the manuscript in response to Reviewers’ comments. We believe that the inclusion of reviewers’ comments has significantly further improved the manuscript.

Responses to Reviewer #3:

I commend the authors on an excellent review of an emerging molecular mechanism of considerable scope. The writing is thorough, which covers the basic details and specifications of circRNAs, to their relationship with miRNAs, and to their plausibility as biomarkers/therapeutic targets.

I have a few minor comments:

Comment: Line 45 – authors could expand on the essential role in synaptic and neuronal functions.

Response: The sentence was deleted.

Comment: Line 56 – consider rewording this line.

Response: The sentence was deleted.

Comment: Intro – The behaviour of circRNAs acting as sponge for miRNAs can be covered and expanded upon first in the introduction.

Response:  We thank the Reviewer for the comments. As suggested, we have added the behaviour of circRNAs acting as sponge for miRNAs to Introduction. Lines 40-43 “Circular RNAs contain miRNA response elements, which bind to miRNAs and compete for miRNA-binding sites. Consequently, circRNAs function as intracellular competitive endogenous RNA (ceRNA) by antagonizing miRNA function, which plays a key role in lung development and disease.”

Comment: Figure 1 – the lariat formation could be labelled in the figure

Response:  The lariat-induced formation has been labelled in Figure 1.

Comment: Line 112 – mention ribonuclease and the relative stability of circRNA in the body compared with RNA.

Response:  As suggested, we made the modification. Lines 108-110 “their resistance to fluid absorption and ribonuclease (RNase) degradation leads to the relative stability of circRNA in the body compared to linear RNA.”

Comment: Line 118  - Seems sensible to reorder the species with humans coming last, i.e rodent, pig, human..

Response:  Done.

Comment: Line 120 – (snRNP) expand on first use

Response: Done.

Comment: Line 181 – authors can reword this sentence to include something of the like “dysregulation, imbalance and abnormal lung development can induce…..”

Response: As suggested, we made the modification. Lines 270-272 “the dysregulated proteins involved in the abnormal lung development can induce inflammation and suppress the growth of alveoli and blood vessels, potentially leading to developmental and pediatric lung diseases”

Comment: Heading 2.3 – This topic seems to be of extra significance. Perhaps this paragraph can be expanded to include more information at hand.

Response: As suggested, we have added more information to Part 2.3. Lines 185-188 “To be noted, circRNAs' dynamic tertiary structure may be affected by different cell types, tissues, and developmental stages, which can affect their ability to bind to various proteins. Circular RNAs can display a variety of functions as a result of different circRNA-RBP interactions [66].”

Response: Line 304 – expand Rian on first use.   

Comment: Done.